

# Investigating the influence of eco-friendly approaches on saline soil traits and growth of common bean plants (*Phaseolus vulgaris* L.)

Tamer Khalifa[1], Nasser Ibrahim Abdel-Kader[2], Mohssen Elbagory[3], Mohamed ElSayed Ahmed[4], Esraa Ahmed Saber[2], Alaa El-Dein Omara[5] and Rehab Mohamed Mahdy[4]

[1] Soil Improvement and Conservation Research Department, Soil, Water, and Environment Research Institute (SWERI), Agriculture Research Center (ARC), Giza, Egypt
[2] Soil and Water Department Faculty of Agriculture, Tanta University, Tanta, Egypt
[3] Department of Biology, Faculty of Science and Arts, King Khalid University, Assir, Mohail, Saudi Arabia
[4] Horticulture Department, Faculty of Agriculture, Tanta University, Tanta, Egypt
[5] Soil Microbiology Research Department, Soil, Water, and Environment Research Institute (SWERI), Agriculture Research Center (ARC), Giza, Egypt

Corresponding author
Tamer Khalifa, tamerkhalifa1985@gmail.com

## ABSTRACT

Soil salinization significantly impacts agricultural lands and crop productivity in the study area. Moreover, freshwater scarcity poses a significant obstacle to soil reclamation and agricultural production. Therefore, eco-friendly strategies must be adopted for agro-ecosystem sustainability under these conditions. A study conducted in 2022 and 2023 examined the interaction effects of various soil mulching materials (unmulched, white plastic, rice straw, and sawdust) and chitosan foliar spray application (control, 250 mg L$^{-1}$ of normal chitosan, 125 mg L$^{-1}$ of nano chitosan, and 62.5 mg L$^{-1}$ of nano chitosan) on the biochemical soil characteristics and productivity of common beans in clay-saline soil. Higher organic matter, available nutrient content, and total bacteria count in soils were found under organic mulching treatments (rice straw and sawdust). In contrast, the white plastic mulching treatment resulted in the lowest values of soil electrical conductivity (EC) and the highest soil water content. Conversely, chitosan foliar spray treatments had the least impact on the chemical properties of the soil. Plants sprayed with 62.5 mg L$^{-1}$ of nano chitosan exhibited higher chlorophyll content, plant height, fresh weight of shoots and roots, seed yield, and nutrient content compared to other chitosan foliar spray applications. All treatments studied led to a significant reduction in fungal communities and Na% in plants. The combined effect of organic mulch materials and foliar spray application of 62.5 mg L$^{-1}$ nano chitosan appeared to enhance biochemical saline soil properties and common bean productivity.

## INTRODUCTION

Soil salinization is recognized as a significant global environmental and socioeconomic challenge, which may exacerbate with climate change (*Dawood et al., 2022*). Furthermore, it has raised concerns about the sustainability of agricultural production (*Morton et al., 2019*). Globally, soil salinization affects approximately one billion hectares of cultivated land, with its expansion attributed to human activities and climate change (*Hopmans et al., 2021*). In the Kafr El-Sheikh Governorate, North Nile Delta of Egypt, soil salinization poses a problem for nearly 56% (151.05 hectares) of the total cultivated area, as reported by *Aboelsoud et al. (2022)*. Additionally, Egypt has experienced freshwater shortages in recent years (*Sofy et al., 2021*). Salt accumulation in arid climates, such as Egypt, can be attributed to various factors, including soil texture, seawater intrusion (*Stavi, Thevs & Priori, 2021*), irrigation dependency on wastewater (*Corwin, 2021*), and inefficient drainage networks. These factors contribute to soil quality degradation, leading to disruptions in fertility, decreased microbial activity, and compromised soil structure (*Al-Suhaibani et al., 2021*; *Devkota et al., 2022*). Consequently, the high accumulation of $Na^+$ and $Cl^-$ can induce osmotic stress (*Van Zelm, Zhang & Testerink, 2020*), reduce nutrient uptake (*Hasanuzzaman & Fujita, 2022*), and cause significant damage to proteins, nucleic acids, membrane lipids (*Guo et al., 2022*), and chloroplasts (*Hameed et al., 2021*).

Cultivated globally for its edible green pods and dry seeds, common bean (*Phaseolus vulgaris* L.) is a self-pollinated, diploid legume crop within the Fabaceae family (*Priya & Manickavasagan, 2020*; *Jan et al., 2021*). However, this crop is highly sensitive to salt stress, with significant declines in both morphological attributes and yield (*Garcia et al., 2019*; *El-Beltagi et al., 2023*). Therefore, managing salinity dynamics requires the adoption of eco-friendly strategies to improve soil characteristics and increase crop productivity, particularly under freshwater limitations.

Among eco-friendly strategies, soil mulch is known to have potential in lower soil evaporation, thus enhancing the soil water content and reducing salt buildup in the soil (*Alkharabsheh et al., 2021*). Soil mulching can involve the use of various materials, including organic options like straws, husks, and grasses, as well as inorganic materials such as plastic, non-woven fabrics, and paper films (*Kader et al., 2017*). The previous studies indicated that the plastic mulch materials have a stronger positive impact on soil physicochemical properties and yield than organic materials (*Haque, Jahiruddin & Clarke, 2018*; *Qu et al., 2019*; *Amare & Desta, 2021*; *Uju et al., 2022*). However, the agro-system environment can be adversely affected by its low recycling rate (*Yang et al., 2020*; *Dewi et al., 2024*). To address this challenge, employing organic mulch materials emerges as a promising solution due to their cost-effectiveness and environmental friendliness (*Zhao et al., 2016a*; *Zhao et al., 2016b*). Incorporating straw into the soil can indeed offer several benefits, such as inhibiting salt accumulation and increasing soil water content (*Song et al., 2019*; *Xie et al., 2020*; *Li et al., 2023*), enhance the accumulation of organic carbon in the soil (*Wang et al., 2021*), enhance microbial activity (*Zhao et al., 2016a*; *Zhao et al., 2016b*; *Fu et al., 2020*), and soil nutrients (*Pi et al., 2017*; *Iqbal et al., 2020*). However, there

is comparatively limited research on the effects of different mulch materials on common bean crops and traits in saline soil (*Kwambe et al., 2015*; *El-Wahed et al., 2017*).

The use of biostimulants is another alternative to mitigate the salinity effect on plants, by enhancing plant growth and crop productivity (*Cataldo, Fucile & Mattii, 2022*; *Fawzy et al., 2023*). This aligns with established strategies for enhancing plant growth and productivity. Chitosan, a natural polysaccharide biopolymer, is derived from the deacetylation of chitin ($\beta$-(1→4)-linked N-acetyl-d-glucosamine and D-glucosamine). Chitin can be sourced from natural waste, such as crab and shrimp shells, or from bacterial fermentation processes (*Riaz Rajoka et al., 2019*). Chitosan's aqueous solubility typically increases with decreasing molecular weight and increasing acidity (*Seenuvasan, Sarojini & Dineshkumar, 2020*). Since chitosan is an inexpensive material that is eco-friendly and has low toxicity, it has gained considerable attention as a potential biological resource for sustainable production (*Li et al., 2020*).

Nanotechnology has found extensive application in various aspects of plant improvement, where nanoparticles are increasingly replacing bulk materials (*Azameti & Imoro, 2023*). Recently, nano chitosan has garnered significant attention for its capacity to produce environmentally friendly and highly functional materials. Its molecular and nanostructures are influenced by its physical properties (*Tokatlı& Demirdöven, 2020*). These advantages, such as interface and surface effects, combined with their small size, render them more effective than conventional chitosan (*Divya & Jisha, 2018*; *Lee et al., 2023*). Several studies have reported that chitosan or its nanoparticles play a crucial role in alleviating plant oxidative stress and promoting crop productivity under salinity stress. For instance, *Hidangmayum et al. (2019)* found that they can alleviate the harmful effects of salinity stress on plants by increasing the photosynthetic rate, and enhancing the production of organic acids, sugars, and amino acids. They also upgrade several defensive genes in plants, such aspathogenesis-related (*Krupa-Małkiewicz & Fornal, 2018*); regulating cellular osmotic pressure to increase water use efficiency, mineral nutrient uptake, and photosynthesis while reducing oxidative stress (*Zhang et al., 2021*; *Hidangmayum & Dwivedi, 2022*; *Ji et al., 2022*; *Balusamy et al., 2022*); increasing proline levels (*Aazami et al., 2023*); enhancing the activity of antioxidant enzymes and reducing malondialdehyde levels (*Attia et al., 2021*), and modifying antioxidant activities, ion homeostasis, and melatonin levels (*Tabassum et al., 2024*). Thus far, research on utilizing chitosan or its nanoparticles to boost the yield of common beans remains limited. Moreover, little attention has been given to examining potential interaction effects between mulching materials and chitosan or its nanoparticles, particularly in the context of foliar spray applications.Therefore, this research aims to understand the impact of different soil mulching materials combined with a foliar spray application of chitosan or its nanoparticles on common bean productivity and the biochemical characteristics of saline soil.

# MATERIALS & METHODS

## Experimental site

The experiment took place at the Soil Improvement and Conservation Greenhouse of the Sakha Agricultural Research Station, located at coordinates 30°56′ 53″E, 31°05′38″N, with

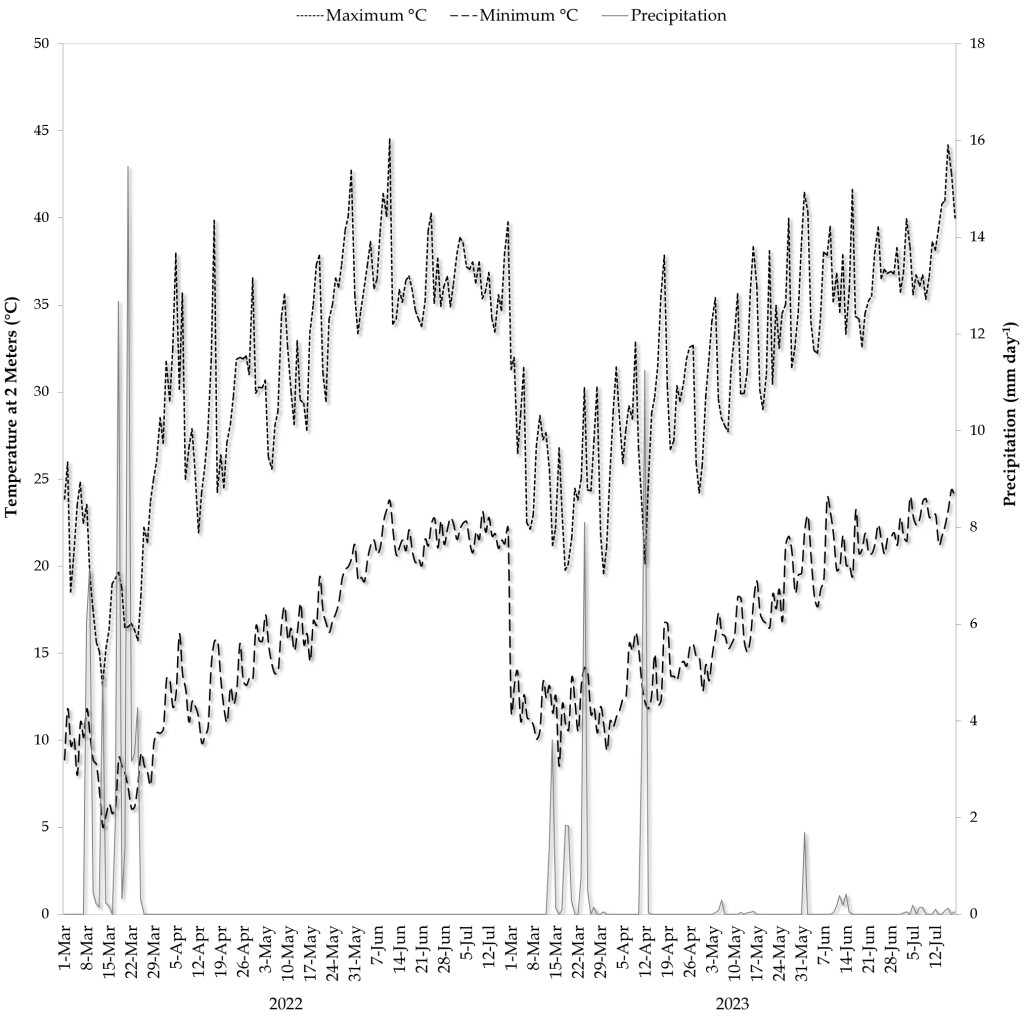

**Figure 1** Daily of temperature and rainfall at the cultivation period of common bean crop (2022 and 2023).

an elevation of approximately 6 m above sea level, in the Kafr El-Sheikh Governorate, North Nile Delta of Egypt. It was conducted during the first week of March in both 2022 and 2023.

Data on daily rainfall and temperature during the cultivation periods of 2022 and 2023 were collected from the weather station installed at the experimental site. The average annual precipitation ranged from 62.21 to 37.84 mm, with temperatures averaging 23.2 and 24.3 °C, respectively (Fig. 1).

The common bean (*Phaseolus vulgaris* L.) cv. Giza 6 seeds were obtained from the Horticulture Research Department at the Agricultural Research Station in Kafr El-Sheikh Governorate. White plastic mulch (30 μm) was sourced from the Arasya Plastic Company

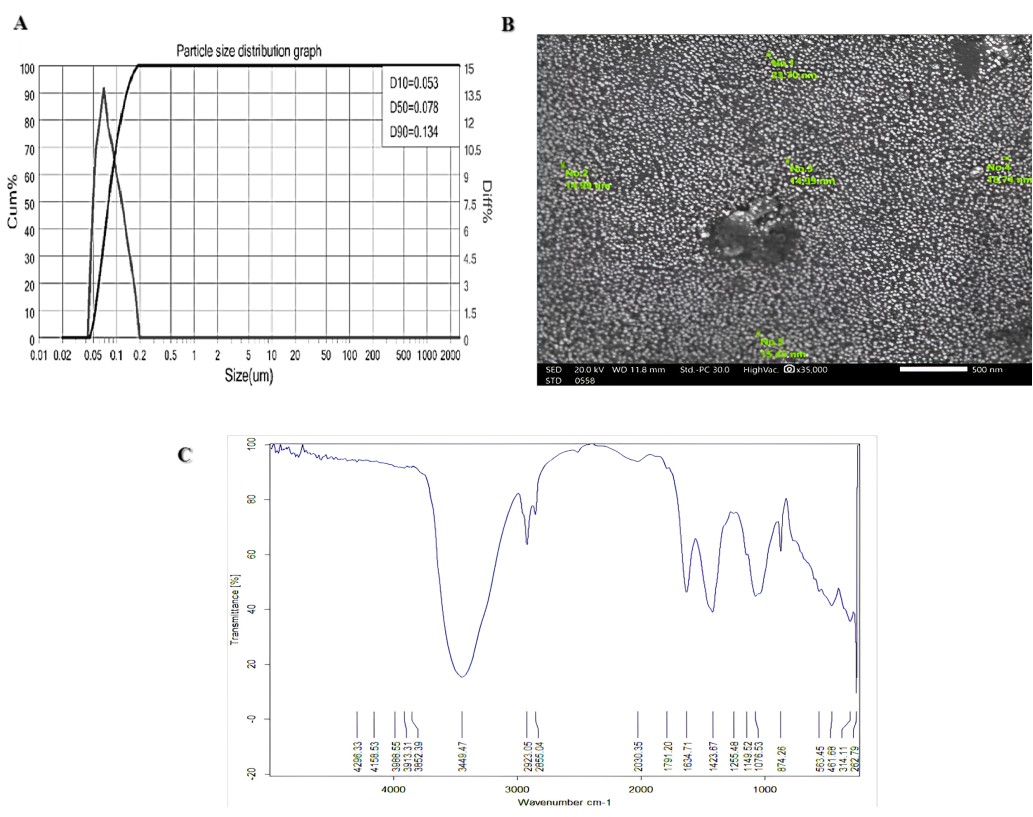

**Figure 2** Characterization of nano chitosan (A) particle size distribution graph, (B) scanning electron microscope (SEM), and (C) the functional groups.

in Heliopolis, Cairo Governorate, while rice straw and sawdust were collected from a local farmer and wood machinery in Kafr El-Sheikh Governorate.

## Synthesis and characterization of nano chitosan

Chitosan ($C_6 H_{11} NO_4$)n was procured from Chitosan Egypt Company, extracted from shrimp shells, with a deacetylation degree of 90%, purity of 95%, and a molecular weight of less than 100 cP. The preparation of nano chitosan followed a method similar to that described by *Sari, Abraha & Suharyadi (2019)*. The ball milling method was employed for 60 min to achieve nanosized particles, followed by drying for 10 min. Particle size analysis was conducted using Bettersize (2600 Wet), and scanning electron microscopy (SEM) images were captured using a JEOL microscope operating at 20 kV at the Nano Research Laboratory of Alexandria University. The chemical structure of nano chitosan was analyzed using a Fourier-transform infrared (FTIR) spectrometer (APD2000Pro) at the Central Laboratory of Tanta University. Figure 2 presents the characterization of nano chitosan.

The average particle size distribution of chitosan nanoparticles was determined to be 86.4 nm (Fig. 2A), indicating nanoscale dimensions. SEM images revealed the formation of spherical nanoparticles (Fig. 2B), confirming the desired morphology. Furthermore,
the functional groups of nano chitosan were identified using FTIR spectra (Fig. 1C). The stretching vibration of O-H, methylene and methyl, C =O, and the C–N stretching, indicating the acetyl group, CN (NH$_2$) stretching as evidence of amine groups, and CN (H$_2$) stretching, were observed at wavenumbers 3,852.39–3,449.47 cm$^{-1}$; 2,923.05–2,855.04 and 2,877 cm$^{-1}$; 1,634.71 and 1,076.53 cm$^{-1}$; 1,423.67 cm$^{-1}$, 1,149.52 cm$^{-1}$, and 1,076.53 cm$^{-1}$, respectively. The presence of strong intermolecular hydrogen bonds in the structure was confirmed by the observed absorptions.

## Experiment setup

Lysimeters were arranged in a randomized block design with three replications. The treatments consisted of two factors: (1) four soil mulching materials, namely un-mulched (UNM), white plastic (WPM), rice straw (RSM), and sawdust (SDM); and (2) four chitosan foliar spray applications: control (distilled water, Ch$_0$); 250 mg L$^{-1}$ of normal chitosan (Ch$_1$); 125 mg L$^{-1}$ of nano chitosan (Ch$_2$); and 62.5 mg L$^{-1}$ of nano chitosan (Ch$_3$). A total of forty-eight lysimeters were utilized, each with dimensions of 0.64 m$^2$ (80 × 80 cm) by 70 cm deep. Each lysimeter was filled with approximately 450 kg of clay soil, with a layer of gravel and sand 10 cm thick at the bottom to aid drainage.

Soil samples were collected from a depth of 30 cm before planting and after harvest, at two time points during the growing seasons of 2022 and 2023. These samples were prepared and analyzed to assess soil characteristics and microbial communities following the protocols outlined by *Page, Miller & Keeney (1982)*, *Nelson & Sommers (1983)*, and *Allen (1958)*.

The soil analysis revealed that the soil texture was classified as heavy clay soil, with clay content at 57.97%, silt at 25.37%, and sand at 18.66%. Additionally, the soil had a high electrical conductivity (EC) of 7.78 dS m$^{-1}$, pH of 8.16, exchangeable sodium percentage (ESP) of 14.76, bulk density of 1.42 mg m$^{-3}$, soil water content of 27.48%, organic carbon content of 0.620 g g$^{-1}$, and available soil nutrients including nitrogen (N) at 23.09 mg kg$^{-1}$, potassium (K) at 204.46 mg kg$^{-1}$, and phosphorus (P) at 11.77 mg kg$^{-1}$. Bacterial and fungal counts were recorded at 5.45 × 10$^3$ CFU and 4.07 × 10$^2$ CFU per gram of dry soil, respectively.

Common beans were planted on the 1st of March in both 2022 and 2023. One ridge per lysimeter was prepared, measuring 60 cm long, 40 cm wide, and 15 cm high. Two seeds were placed in hills 20 cm apart on two sides (6 hills in ridge), later thinned to form a single plant per hill (6 plants per lysimeter). Irrigation water with a pH of 7.08 and an electrical conductivity (ECw) of 2.51 dS m$^{-1}$ was used, and irrigation was adjusted to a depth of 70 cm.

One week after sowing in both seasons, 30-micron white plastic mulch, a five cm thick layer of rice straw, and sawdust were applied to the ridges.

Foliar spraying with chitosan or its nanoparticles commenced 15 days after planting and was repeated twice at intervals of 30 and 45 days in both seasons. Different doses of chitosan or its nanoparticles (250, 125, and 62.5 mg L$^{-1}$) were dissolved in 800 ml of distilled water containing 1% acetic acid. The solution was stirred continuously until

completely dissolved, then topped up to one liter. Finally, the solution was alkalized to pH 6 using a 1 M NaOH solution (*Van, Minh & Anh, 2013*).

Fertilization and agricultural practices followed common bean cultivation practices in the North Nile Delta region of Egypt (49 kg N ha$^{-1}$ as ammonium sulfate, 71.4 kg P ha$^{-1}$ as calcium super phosphate, and 57.1 kg K ha$^{-1}$ as potassium sulfate, in two equal doses). Upon reaching full maturity, the plants were harvested on July 18th of both seasons.

## Growth characters and seed yield of common beans analysis

After 60 days of planting, samples were collected from all plants in each experimental unit in both growing seasons (576 samples) for various measurements during the growing seasons of 2022 and 2023. The measurements included:

I. Chlorophyll content in flag leaves (top leaflet petiole), measured using a Chlorophyll Content Meter (Model CCM-200 plus GPS, USA).
II. Plant height (cm).
III. Shoot and root fresh weight per plant in grams (totaling 288 samples), randomly sampled from three plants per experimental unit during both seasons.

Upon reaching full maturity, the remaining three plants from each experimental unit during both seasons, totaling 288 samples, were weighed to determine seed yield (kg ha$^{-1}$). Additionally, plant samples were dried, ground, and wet-digested following the method described by *Wolf (1982)*. Chemical analysis was then conducted to determine the percentages of nitrogen (N), phosphorus (P), potassium (K), and sodium (Na) according to the standard methods outlined by *Motsara & Roy (2008)*.

## Data analysis and processing

The analysis of variance (ANOVA) for the two-way factorial design can be conducted using the **aov**() function in R software for Windows (ver. 4.3.1) and **Tukey HSD test** to conduct multiple comparisons between different treatments (*R Core Team, 2010*). Differences between treatment interactions are denoted with ggbarplot packages; principal component analysis (PCA) with Factoextra packages; and Visualize Correlation Matrix with corrplot packages (*Wei & Simko, 2021*; *Kassambara & Mundt, 2023*).

## RESULTS

The results of a two-way ANOVA show soil mulch, chitosan foliar spray treatments, and co-applications affect the investigated soil characteristics across two seasons (2022–2023), as depicted in Table 1.

## Soil characteristics

The investigated soil characteristics included soil water content (SWC), soil electrical conductivity (EC), soil organic carbon (SOC), available soil nitrogen (Ava. N), phosphorus (Ava. P), and potassium (Ava. K). Compared to the unmulched treatment, white plastic mulch (WPM) increased SWC by 22.98% and decreased soil EC by 10.37% on average across the growing seasons (Table 2). The use of organic soil mulch increased the average levels of SOC, Ava. N, P, and K in both seasons (Table 2). SOC content increased by 0.159

Khalifa et al. (2024), *PeerJ*, DOI 10.7717/peerj.17828

**Table 1** Results of two-way ANOVA of investigated soil characteristics under the main and co applications effects of soil mulch and chitosan foliar spray applications in two seasons.

| Parameters | Treatments | Season | 2022 | | | 2023 | | |
|---|---|---|---|---|---|---|---|---|
| | | Df | Sum Sq | Mean Sq | Pr(>F) | Sum Sq | Mean Sq | Pr(>F) |
| SWC (%) | Mulching | 3 | 322.4 | 107.5 | <0.001*** | 350.3 | 116.8 | <0.001*** |
| | Chitosan foliar spray | 3 | 2.6 | 0.87 | 0.667 | 3.3 | 1.09 | 0.856 |
| | Interaction | 9 | 0.4 | 0.05 | 1.000 ns | 0.5 | 0.05 | 1.000 ns |
| | Residuals | 30 | 39.1 | 1.31 | | 111.99 | 3.73 | |
| EC (dS m$^{-1}$) | Mulching | 3 | 2.35 | 0.78 | <0.001*** | 5.85 | 1.9489 | <0.001*** |
| | Chitosan foliar spray | 3 | 0.45 | 0.15 | 0.0029** | 0.354 | 0.1181 | <0.001*** |
| | Interaction | 9 | 0.029 | 0.003 | 0.0294* | 0.029 | 0.0032 | 0.99 ns |
| | Residuals | 30 | 0.152 | 0.005 | | 0.500 | 0.0167 | |
| SOC (g g$^{-1}$) | Mulching | 3 | 0.169 | 0.056 | <0.001*** | 0.236 | 0.0785 | <0.001*** |
| | Chitosan foliar spray | 3 | 0.006 | 0.002 | 0.0272* | 0.0158 | 0.0053 | 0.0104* |
| | Interaction | 9 | 0.003 | 0.0004 | 0.8554 ns | 0.0055 | 0.0006 | 0.8566 ns |
| | Residuals | 30 | 0.022 | 7.20e−04 | | 0.035 | 0.0012 | |
| Ava. N (mg g$^{-1}$) | Mulching | 3 | 607 | 202.3 | <0.001*** | 657.1 | 219.03 | <0.001*** |
| | Chitosan foliar spray | 3 | 12.1 | 4.02 | 0.338 ns | 17.68 | 5.895 | 0.0103* |
| | Interaction | 9 | 0.2 | 0.02 | 1.000 ns | 24.4 | 2.72 | 0.704 ns |
| | Residuals | 30 | 98.84 | 3.295 | | 83.49 | 2.783 | |
| Ava. P (mg g$^{-1}$) | Mulching | 3 | 147.4 | 49.13 | <0.001*** | 135.8 | 45.25 | <0.001*** |
| | Chitosan foliar spray | 3 | 4.04 | 1.35 | 0.294 | 6.8 | 2.27 | 0.0761* |
| | Interaction | 9 | 0.22 | 0.02 | 1.000 ns | 1.67 | 0.19 | 0.9915 ns |
| | Residuals | 30 | 30.35 | 1.012 | | 26.98 | 0.899 | |
| Ava. K (mg g$^{-1}$) | Mulching | 3 | 4,555 | 1,519 | <0.001*** | 47,293 | 15,764 | <0.001*** |
| | Chitosan foliar spray | 3 | 11 | 4 | 0.894 ns | 18 | 66 | 0.722 ns |
| | Interaction | 9 | 5 | 1 | 1.000 ns | 5 | 17 | 1.000 ns |
| | Residuals | 30 | 513.2 | 17.11 | | 434.34 | 14.48 | |
| TBC (log CFU g$^{-1}$) | Mulching | 3 | 27.59 | 9.196 | <0.001*** | 34.76 | 11.586 | <0.001*** |
| | Chitosan foliar spray | 3 | 3.38 | 1.126 | <0.001*** | 6.22 | 2.073 | 0.0339* |
| | Interaction | 9 | 0.51 | 0.056 | 0.0126* | 2.11 | 0.234 | 0.941 ns |
| | Residuals | 30 | 0.580 | 0.019 | | 17.92 | 0.597 | |
| TFC (log CFU g$^{-1}$) | Mulching | 3 | 8.953 | 2.9844 | .00673** | 23.724 | 7.908 | <0.001*** |
| | Chitosan foliar spray | 3 | 2.411 | 0.8037 | 0.288 ns | 4.495 | 1.498 | 0.0016** |
| | Interaction | 9 | 0.072 | 0.008 | 0.000 ns | 0.334 | 0.037 | 0.9999 ns |
| | Residuals | 30 | 15.66 | 0.522 | | 16.72 | 0.558 | |

**Notes.**

SWC, soil water content; EC, soil salinity; SOC, soil organic carbon content; Ava. N, P, and K, available N, P and K content; TBC, total bacteria count; TFC, total fungi count.

*,**,***Significant codes: *** 0.001, ** 0.01, * 0.05, and **ns** not statistically significant.

**Table 2  Statistical results for the main effects of mulching treatment and chitosan foliar spray applications on soil parameters in two seasons (2022–2023).**

| Season Parameters | SWC (%) | EC (dS m$^{-1}$) | SOC (g g$^{-1}$) | Ava. N (mg g$^{-1}$) | 2022 Ava. P (mg g$^{-1}$) | Ava. K (mg g$^{-1}$) | TBC (log CFU g$^{-1}$) | TFC (log CFU g$^{-1}$) |
|---|---|---|---|---|---|---|---|---|
| Mulching materials | | | | | | | | |
| UNM | 27.14c | 7.77a | 0.598d | 27.26c | 12.57b | 201.19d | 5.92d | 3.61a |
| WPM | 33.54a | 7.15c | 0.643c | 33.61b | 13.68b | 256.36c | 6.96c | 3.06ab |
| RSM | 29.63b | 7.51b | 0.695b | 36.33a | 16.89a | 266.13b | 7.45b | 3.15ab |
| SDM | 32.94a | 7.40b | 0.757a | 35.50ab | 16.10a | 283.55a | 7.98a | 2.40b |
| Chitosan foliar spray | | | | | | | | |
| Ch$_0$ | 31.15 | 7.61a | 0.659b | 32.32 | 15.30 | 251.03 | 6.64c | 3.43 |
| Ch$_1$ | 30.87 | 7.47b | 0.669ab | 33.35 | 14.74 | 251.86 | 7.10b | 3.03 |
| Ch$_2$ | 30.72 | 7.41b | 0.676ab | 33.44 | 14.66 | 252.09 | 7.26ab | 2.92 |
| Ch$_3$ | 30.51 | 7.35c | 0.690a | 33.59 | 14.55 | 252.26 | 7.31a | 2.85 |
| **Season** | | | | | 2023 | | | |
| Mulching materials | | | | | | | | |
| UNM | 27.89b | 7.60a | 0.602d | 27.56c | 12.38c | 201.34d | 5.96c | 3.31a |
| WPM | 34.13a | 6.63d | 0.660c | 34.60b | 14.19b | 257.31c | 7.23b | 2.47b |
| RSM | 29.16b | 7.16b | 0.732b | 36.90a | 16.82a | 268.91b | 7.49ab | 2.31b |
| SDM | 33.55a | 7.01c | 0.786a | 36.17ab | 15.82a | 284.63a | 8.33a | 1.33c |
| Chitosan foliar spray | | | | | | | | |
| Ch$_0$ | 31.58 | 7.23a | 0.668b | 32.77b | 15.42 | 252.05 | 6.66c | 2.86a |
| Ch$_1$ | 31.22 | 7.12ab | 0.691ab | 34.03a | 14.74 | 253.11 | 7.28ab | 2.33b |
| Ch$_2$ | 31.08 | 7.05bc | 0.705ab | 34.14a | 14.61 | 253.31 | 7.45ab | 2.15b |
| Ch$_3$ | 30.86 | 7.00c | 0.716a | 34.29a | 14.43 | 253.71 | 7.62a | 2.08b |

**Notes.**
SWC, soil water content; EC, soil salinity; S.O.C, soil organic carbon content; Ava. N, P, and K, available N, P and K content; TBC, total bacteria count; TFC, total fungi count; UNM, un-mulched; WPM, white plastic mulching; RSM, rice straw mulching; SDM, sawdust mulching; Ch$_0$, Control (distilled water); Ch$_1$, 250 mg L$^{-1}$ of normal chitosan; Ch$_2$, 125 mg L$^{-1}$ of nano chitosan; Ch$_3$, 62.5 mg L$^{-1}$ of nano chitosan.
The means that do not share a common letter are considered significantly different at 0.05.

and 0.184 g g$^{-1}$ under sawdust mulch (SDM) treatment, followed by rice straw mulch (RSM) treatment, which increased by 0.097 and 0.130 g g$^{-1}$ compared to unmulched soil in both seasons, respectively. We did not find statistically significant differences in SWC and soil available nutrients except for Ava. N and P in 2023 with chitosan foliar spray application (Table 1). Chitosan foliar spray applications had an impact on soil EC in both 2022 and 2023 (Table 1). Moreover, treatment Ch$_3$ had a lower value of soil EC compared to other foliar application treatments and a higher value of SOC (0.690 and 0.716 g g$^{-1}$) in both seasons, respectively (Table 2). Regarding the interaction effect, the co-application of soil mulch treatment and chitosan foliar spray applications did not show significance in the investigated soil characteristics during the two growing seasons (Table 1); only significant differences between treatments were observed in soil EC in 2022 (Fig. 3). The co-application of WPM with Ch$_3$ resulted in a lower value of soil EC (7.04 dS m$^{-1}$).

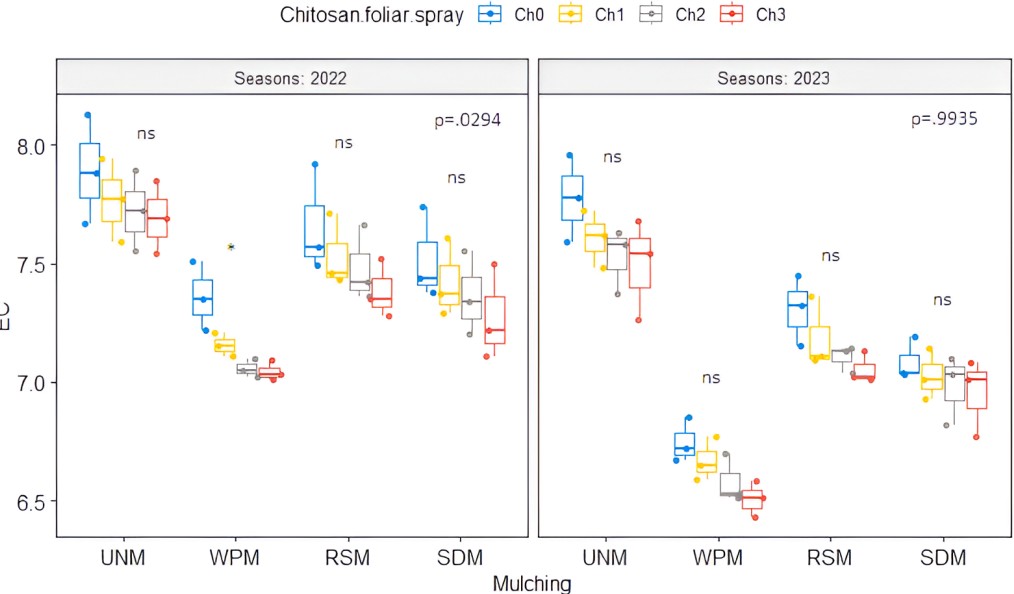

**Figure 3  Soil EC (dS m⁻¹) for the interaction affect of mulching treatment with chitosan foliar spray applications in 2022 and 2023.** UNM, un-mulched; WPM, white plastic mulching; RSM, rice straw mulching; SDM, sawdust mulching, $Ch_0$, Control (distilled water); $Ch_1$, 250 mg $L^{-1}$ of normal chitosan; $Ch_2$, 125 mg $L^{-1}$ of nano chitosan; $Ch_3$, 62.5 mg $L^{-1}$ of nano chitosan. '*' is significant, and 'ns' is insignificant at $p < 0.05$. Mean value ± standard error.

## Soil microbial community

There was a significant change in the investigated soil microbial community under different soil mulch treatments (Table 1). The sawdust mulch (SDM) treatment exhibited significantly greater bacterial communities (7.98 and 8.33 log CFU $g^{-1}$) in both seasons, respectively, but no differences were found compared to rice straw mulch (RSM) in 2022 (Table 2). Additionally, a lower fungi community was observed under SDM compared to unmulched (UNM) soil in both seasons (Table 2). The results also showed that insignificant differences were found between soil mulch treatments.

The effect of foliar spray applications of chitosan or its nanoparticles, compared with the control treatment ($Ch_0$), resulted in a statistically significant increase in bacterial communities in 2022 and 2023 (Table 2). Compared to the control treatment, whole chitosan or its nanoparticles foliar spray application treatment decreased the average fungi community in the growing seasons of 2022 and 2023. These reductions were 18.53%, 24.83%, and 27.27% for $Ch_1$, $Ch_2$, and $Ch_3$, respectively, in 2023.

Regarding the co-application effect, the data in Table 1 indicate that the only significant differences in bacterial communities between treatments were observed in 2022. The SDM with two doses of foliar spraying by nano chitosan ($Ch_2$ and $Ch_3$) had a higher value of bacterial communities (8.11 and 8.13 log CFU $g^{-1}$, respectively) in 2022 (Fig. 4). The fungi community was insignificantly different across treatments in 2022 and 2023.

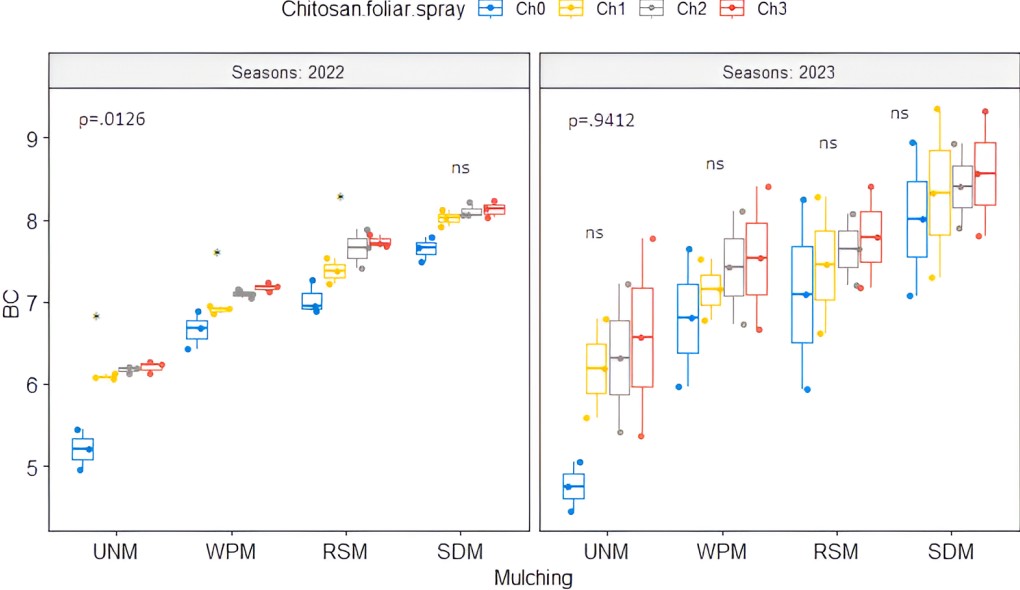

**Figure 4  Bacteria community (log CFU g$^{-1}$) for the interaction effect of mulching treatment with chitosan foliar spray applications in 2022 and 2023.** UNM, un-mulched; WPM, white plastic mulching; RSM, rice straw mulching; SDM, sawdust mulching; Ch$_0$, Control (distilled water); Ch$_1$, 250 mg L$^{-1}$ of normal chitosan; Ch$_2$, 125 mg L$^{-1}$ of nano chitosan; Ch$_3$, 62.5 mg L$^{-1}$ of nano chitosan. '*' is significant, and 'ns' is insignificant at $p < 0.05$. Mean value ± standard error.

## Morphological attributes and seed yield of common beans

The two-way ANOVA results showed that the main soil mulch treatments and chitosan foliar spray applications significantly affected the investigated morphological attributes and seed yield of common beans in both seasons (Table 3). Co-application results in the investigated morphological attributes and seed yield of common beans revealed that insignificant differences were found in both seasons.

Different soil mulch materials showed significantly higher chlorophyll content and plant height compared with the unmulching treatment (UNM) during both seasons, but no differences in the values of chlorophyll content and plant height were found between mulching treatments in 2022 (Table 4). In the 2023 season, RSM had higher values of chlorophyll content and plant height than the soil mulch treatment. Relative to UNM, the plant height was decreased by 13.44%, 16.46%, and 13.60% in plants mulched by WPM, RSM, and SDM, respectively. Organic mulching materials gradually increased the fresh weight of shoot and root compared to WPM in both seasons (Table 4), with the minimum values recorded with the unmulched treatment (UNM). The seed yield increased under all soil mulch treatments compared to UNM. Moreover, insignificant differences in seed yield were observed between WPM, RSM, and SDM in both seasons (Table 4).

Furthermore, chlorophyll content was substantially enhanced with Ch$_3$ application compared to Ch$_0$ in 2022 and 2023 (Table 4). The reduction in plant height in control plants (Ch$_0$) were 18.99%, 20.07%, and 20.98% compared with Ch$_1$, Ch$_2$, and Ch$_3$,

**Table 3  Results of two-way ANOVA of investigated common bean parameters under the main and co applications effects of soil mulch and chitosan foliar spray applications in two seasons.**

| Parameters | Treatments | Season | 2022 | | | 2023 | | |
|---|---|---|---|---|---|---|---|---|
| | | Df | Sum Sq | Mean Sq | Pr(>F) | Sum Sq | Mean Sq | Pr(>F) |
| Chlorophyll (mg g$^{-1}$ f.w) | Mulching | 3 | 270.6 | 90.2 | <0.001*** | 304.9 | 101.7 | <0.001*** |
| | Chitosan foliar spray | 3 | 378.9 | 126.3 | <0.001*** | 404.4 | 134.8 | <0.001*** |
| | Interaction | 9 | 11.7 | 1.3 | 0.617 ns | 9.4 | 1.04 | 0.657 ns |
| | Residuals | 30 | 31.3 | 1.04 | | 28.2 | 0.94 | |
| Plant height (cm) | Mulching | 3 | 120.7 | 40.2 | <0.001*** | 143.97 | 47.99 | <0.001*** |
| | Chitosan foliar spray | 3 | 207.5 | 69.2 | <0.001*** | 213.04 | 71.01 | <0.001*** |
| | Interaction | 9 | 1.83 | 0.20 | 0.969 ns | 2.54 | 0.28 | 0.883 ns |
| | Residuals | 30 | 4.1 | 0.14 | | 3.44 | 0.12 | |
| Shoot FW (g plant$^{-1}$) | Mulching | 3 | 248.5 | 82.83 | <0.001*** | 300.49 | 100.16 | <0.001*** |
| | Chitosan foliar spray | 3 | 262.5 | 87.51 | <0.001*** | 290.01 | 96.67 | <0.001*** |
| | Interaction | 9 | 0.79 | 0.09 | 0.998 ns | 0.83 | 0.09 | 1 ns |
| | Residuals | 30 | 4.52 | 1.51e−01 | | 23.39 | 0.78 | |
| Root FW (g plant$^{-1}$) | Mulching | 3 | 50.61 | 16.87 | <0.001*** | 84 | 28 | <0.001*** |
| | Chitosan foliar spray | 3 | 83.3 | 27.767 1 | <0.001*** | 99.9 | 33.3 | <0.001*** |
| | Interaction | 9 | 0.39 | 0.043 | 0.996 ns | 0.57 | 0.06 | 0.984 ns |
| | Residuals | 30 | 6.25 | 0.21 | | 6.90 | 0.23 | |
| Seed yield (Kg ha$^{-1}$) | Mulching | 3 | 2,644 | 8,812 | <0.001*** | 2,682 | 8,940 | <0.001*** |
| | Chitosan foliar spray | 3 | 1,660 | 5,534 | <0.001*** | 1,800 | 5,999 | <0.001*** |
| | Interaction | 9 | 1,166 | 130 | 1 ns | 483 | 53 | 1 ns |
| | Residuals | 30 | 1,565.3 | 5,215.7 | | 1,589.8 | 5,296.3 | |
| N (%) | Mulching | 3 | 5.22 | 1.75 | <0.001*** | 5.642 | 1.88 | <0.001*** |
| | Chitosan foliar spray | 3 | 4.16 | 1.39 | <0.001*** | 1.992 | 0.664 | <0.001*** |
| | Interaction | 9 | 0.15 | 0.017 | <0.001*** | 0.525 | 0.058 | <0.001*** |
| | Residuals | 30 | 0.032 | 0.0011 | | 0.047 | 0.0016 | |
| P (%) | Mulching | 3 | 0.324 | 0.108 | <0.001*** | 0.376 | 0.125 | <0.001*** |
| | Chitosan foliar spray | 3 | 0.473 | 0.158 | <0.001*** | 0.601 | 0.200 | <0.001*** |
| | Interaction | 9 | 0.009 | 0.001 | .0267* | 0.013 | 0.0014 | 0.0017** |
| | Residuals | 30 | 0.0017 | 5.65e−05 | | 0.0012 | 4.09e−05 | |
| K (%) | Mulching | 3 | 5.75 | 1.92 | <0.001*** | 7.524 | 2.508 | <0.001*** |
| | Chitosan foliar spray | 3 | 3.58 | 1.19 | <0.001*** | 2.407 | 0.802 | <0.001*** |
| | Interaction | 9 | 0.22 | 0.025 | <0.001*** | 0.317 | 0.035 | <0.001*** |
| | Residuals | 30 | 0.013 | 0.0005 | | 0.021 | 0.0007 | |
| Na (%) | Mulching | 3 | 0.100 | 0.033 | <0.001*** | 0.113 | 0.038 | <0.001*** |
| | Chitosan foliar spray | 3 | 0.274 | 0.091 | <0.001*** | 0.281 | 0.094 | <0.001*** |
| | Interaction | 9 | 0.027 | 0.003 | <0.001*** | 0.027 | 0.004 | <0.001*** |
| | Residuals | 30 | 0.009 | 0.0003 | | 0.009 | 0.0003 | |

**Notes.**
FW, fresh weight.
Significant codes: '***'0.001, '**'0.01, '*'0.05, and 'ns' not statistically significant.

respectively. In addition, the fresh weight of the shoot was increased by 38.76% and 37.17% with Ch$_3$ application, and the increases in fresh root weight were 42.36% and 44.04% during the seasons of 2022 and 2023, respectively, compared with the control

**Table 4** Statistical results for the main effects of mulching treatment and chitosan foliar spray applications on investigated common bean parameters in two seasons (2022–2023).

| Season Parameters | Chlorophyll | Ph | Shoot FW | Root FW | 2022 SY | N | P | K | Na |
|---|---|---|---|---|---|---|---|---|---|
| | (mg g$^{-1}$ f.w) | (cm) | (g) | (g) | (kg ha$^{-1}$) | (%) | (%) | (%) | (%) |
| Mulching materials | | | | | | | | | |
| UNM | 40.84b | 24.81b | 29.58c | 8.58c | 1,450.44b | 2.65d | 0.461d | 2.06c | 0.458a |
| WPM | 45.88a | 28.14a | 33.98b | 10.20b | 1,981.35a | 3.23c | 0.607c | 2.75b | 0.352b |
| RSM | 46.86a | 28.89a | 35.15a | 11.35a | 2,003.96a | 3.51a | 0.673a | 2.91a | 0.349bc |
| SDM | 46.05a | 28.18a | 34.38ab | 10.70ab | 1,991.04a | 3.39b | 0.647b | 2.89a | 0.358a |
| Chitosan foliar spray | | | | | | | | | |
| Ch0 | 40.23c | 23.92c | 26.37d | 7.98c | 1,543.36c | 2.74d | 0.441d | 2.23d | 0.499a |
| Ch1 | 46.28b | 28.46b | 34.68c | 10.58b | 1,918.59b | 3.14c | 0.588c | 2.60c | 0.384b |
| Ch2 | 45.46ab | 28.72a | 35.44b | 10.91ab | 1,933.92b | 3.36b | 0.655b | 2.81b | 0.337c |
| Ch3 | 47.65a | 28.93a | 36.59a | 11.36a | 2,030.93a | 3.53a | 0.705a | 2.96a | 0.298c |
| Season | | | | | 2023 | | | | |
| Mulching materials | | | | | | | | | |
| UNM | 41.29c | 24.91c | 30.25c | 8.88d | 1,481.24b | 2.76d | 0.525d | 2.09d | 0.435a |
| WPM | 46.36b | 28.42ab | 34.64b | 10.74c | 2,014.03a | 3.28c | 0.686c | 2.74c | 0.323b |
| RSM | 47.88a | 29.37a | 35.70a | 12.50a | 2,040.54a | 3.63a | 0.761a | 3.01b | 0.319b |
| SDM | 46.65b | 28.70b | 34.90a | 11.48b | 2,025.55a | 3.55b | 0.714b | 3.11a | 0.326b |
| Chitosan foliar spray | | | | | | | | | |
| Ch0 | 40.64c | 24.22c | 26.93b | 8.47c | 1,565.80c | 2.98d | 0.494d | 2.38d | 0.471a |
| Ch1 | 46.85b | 28.75b | 36.03a | 11.24b | 1,942.63b | 3.30c | 0.667c | 2.72c | 0.359b |
| Ch2 | 46.48b | 29.09ab | 36.53a | 11.69ab | 1,977.14b | 3.41b | 0.732b | 2.87b | 0.306c |
| Ch3 | 48.21a | 29.33a | 36.94a | 12.20a | 2,075.77a | 3.53a | 0.793a | 2.97a | 0.268c |

**Notes.**

Ph, plant height; FW, shoots and roots fresh weight; SY, seed yield; UNM, un-mulched; WPM, white plastic mulching; RSM, rice straw mulching; SDM, sawdust mulching; Ch$_0$, Control (distilled water); Ch$_1$, 250 mg L$^{-1}$ of normal chitosan; Ch$_2$, 125 mg L$^{-1}$ of nano chitosan; Ch$_3$, 62.5 mg L$^{-1}$ of nano chitosan.
The means that do not share a common letter are considered significantly different at 0.05.

plants (Table 4). Foliar spray application of 62.5 mg L$^{-1}$ of nano-chitosan (Ch$_3$) had higher seed yields (2030.93 and 2075.77 kg ha$^{-1}$) in 2022 and 2023, respectively (Table 4).

### Nutrient concentrations in common bean plants

Based on the two-way ANOVA analysis, the effects of different mulch materials, chitosan spray foliar applications, and their interaction on N, P, and K concentrations in plants were observed (Table 3). The use of organic mulch materials in the soil increased NPK concentrations in plants. Specifically, SDM had a positive effect on reducing Na concentration in plants compared to the unmulched (UNM) treatment in 2022, although no difference in Na% values among mulching treatments was observed in 2023 (Table 4).

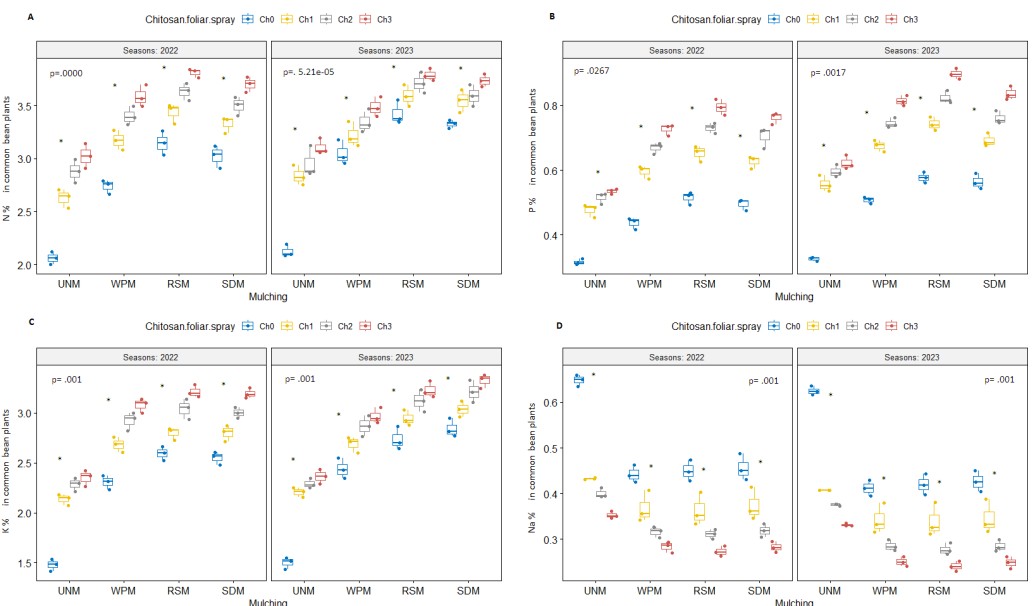

**Figure 5** N, P, N, and Na concentration in common bean plants for the co-application affect of soil mulch treatment with chitosan foliar spray applications in 2022 and 2023. (A) N concentration in common bean plants; (B) P concentration in common bean plants; (C) K concentration in common bean plants; (D) Na concentration in common bean plants. UNM: un-mulched; WPM, white plastic mulching; RSM, rice straw mulching; SDM, sawdust mulching; $Ch_0$, Control (distilled water); $Ch_1$, 250 mg L$^{-1}$ of normal chitosan; $Ch_2$, 125 mg L$^{-1}$ of nano chitosan; $Ch_3$, 62.5 mg L$^{-1}$ of nano chitosan. '*' is significant, and 'ns' is insignificant at $p < 0.05$. Mean value ± standard error.

In comparison to control plants, chitosan spray foliar applications significantly reduced Na% in common bean plants and increased NPK concentrations, particularly with $Ch_3$ applications (Table 4).

Considering the co-application effect, both organic mulch materials and 62.5 mg L$^{-1}$ of nano-chitosan increased NPK concentrations in common bean plants (Figs. 5A, 5B, and 5C), while Na% decreased significantly compared to control plants (Fig. 5D).

## Relationships between the parameters studied of soil and common bean plants

The PCA biplot (Fig. 6) illustrates two dimensions explaining 75.5% of the total variance: Dim1 = 60% and Dim2 = 10.5%. Dim1 indicates a strong negative association between soil EC and total fungi community with soil water content, available N, P, and K, soil organic carbon, total bacteria community, and K% in common bean plants. On the other hand, Dim2 demonstrates a negative correlation between Na% in common bean plants and seed yield, chlorophyll content, plant height, fresh weight of shoot and root, N%, and P% in common bean plants. Differences are more distinctly noticeable for treatments than for seasons. The interaction between un-mulching and different chitosan foliar spray application treatments ($Ch_0$, $Ch_1$, $Ch_2$, and $Ch_3$) exhibited negative variables, such as soil EC, fungi community, and Na% in common bean plants. The correlation results

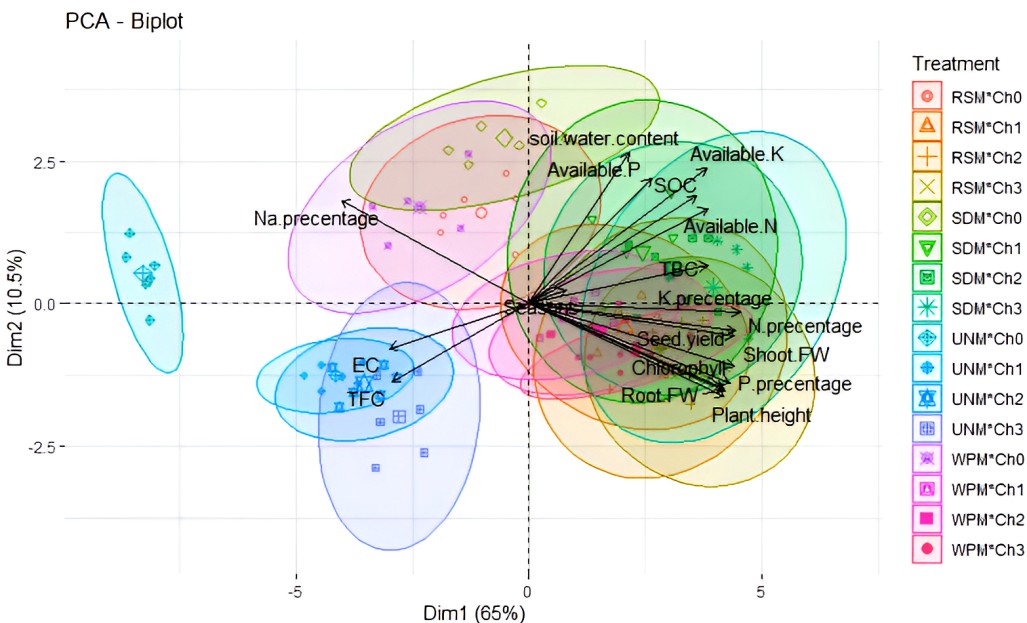

**Figure 6 PCA biplot had response to co-application affect of soil mulch treatment with chitosan foliar spray applications on the parameters studied of soil and common bean plants.** * SWC, soil water content; EC, soil salinity; S.O.C, soil organic carbon content; Ava. N, P, and K, available N, P and K content; TBC, total bacteria count and TFC, total fungi count. UNM, un-mulched; WPM, white plastic mulching; RSM, rice straw mulching; SDM, sawdust mulching; $Ch_0$, Control (distilled water); $Ch_1$, 250 mg $L^{-1}$ of normal chitosan, $Ch_2$, 125 mg $L^{-1}$ of nano chitosan, $Ch_3$, 62.5 mg $L^{-1}$ of nano chitosan.

indicate that improved soil characteristics have a significant influence on common bean productivity.

## DISCUSSION

In this study, we observed that both mulching treatments and nano chitosan foliar spray applications influenced all parameters under investigation, while their interaction had minimal impact, particularly on soil parameters. Furthermore, the year 2023 experienced an overall freshwater shortage, higher temperatures, and lower precipitation compared to 2022. These climatic changes significantly affected soil quality and yield production in the study area. Our findings confirmed that organic mulching materials, such as rice straw or sawdust, led to higher soil organic carbon (SOC), increased availability of soil nutrients, and enhanced bacterial community in the soil during the growing seasons. Additionally, white plastic mulching reduced significantly soil electrical conductivity (EC), while both white plastic and sawdust mulching increased soil water content (SWC).

Moreover, nano chitosan foliar spray application at a concentration of 62.5 mg $L^{-1}$ resulted in elevated chlorophyll content, increased plant height, enhanced fresh weight of shoot and root, and higher nutrient content in common bean plants. The interaction effect contributed to increased bacterial community in soil, enhanced plant height, and higher NPK concentration in common bean plants, attributed to lower soil EC and reduced

sodium uptake by plants. However, there was a decrease in the fungi community due to the use of different mulch materials and chitosan foliar spray applications.

Overall, our study suggests that the use of organic mulch materials (such as rice straw or sawdust) and nano chitosan foliar spray application at a concentration of 62.5 mg L$^{-1}$ could serve as effective and eco-friendly strategies for mitigating the effects of saline soil, because of (1) could enhance soil water content, available N, P, and K, soil organic carbon, total bacteria community lead to decrease on the soil electrical conductivity (EC), fungal communities and sodium uptake by plants. (2) These factors contribute to greater NPK absorption from the soil and lower Na content in common bean plants (3) resulted in enhancements in growth attributes and seed yield of common beans (Fig. 6).

Although the interaction between different soil mulching materials and chitosan foliar spray applications had a statistically low significant impact on biochemical soil parameters, it notably influenced soil electrical conductivity (EC) and total bacteria count in soil. Additionally, there was a general trend towards slightly higher average soil water content, organic carbon, and available nitrogen (N), phosphorus (P), and potassium (K), while the fungi community in soil showed a lower abundance.

The observed effect of plastic mulch aligns with findings from previous studies, which have consistently reported a decrease in soil electrical conductivity (EC) alongside an increase in soil water content (SWC) (Haque, Jahiruddin & Clarke, 2018; Qu et al., 2019; Amare & Desta, 2021; Uju et al., 2022; Dewi et al., 2024). This relationship is logical since higher soil moisture levels lead to reduced soil evaporation and increased water content, facilitating salt leaching and subsequently lowering salt accumulation in the soil. Additionally, research has highlighted the benefits of mulching in arid regions prone to salinity issues, as it mitigates the capillary rise process, further aiding in salt management (Zhao et al., 2016a; Zhao et al., 2016b).

The increase of SOC and the availability of soil nutriments can be attributed to the elevated organic matter content in the soil resulting from the use of organic mulching materials. This finding is consistent with research by Paunović, Milinković & Pešaković (2020), which demonstrated that sawdust mulching treatments led to an increase in soil organic carbon levels. Moreover, the rise in organic matter content may be linked to the decomposition rate of organic mulching materials, which enhances organic carbon accumulation (Fu et al., 2020). The increased availability of nutrients can be attributed to heightened biological activity, facilitating organic matter mineralization processes (Pi et al., 2017).

Interestingly, the impact of different mulching materials on the soil microbial community was somewhat contradictory. While the bacterial community flourished due to the buildup of organic carbon in the soil, the fungal community experienced a decline as a consequence of the mulching treatments. These results are in line with those reported by Iqbal et al. (2020).

The results indicated a positive impact of chitosan foliar application on soil salinity and carbon content. This could be attributed to its role in facilitating root penetration through soil layers, thereby aiding in the leaching of soil salts. Additionally, the presence of root residues in the soil contributed to increased organic matter, subsequently enhancing soil

organic carbon levels and bacterial populations. Chitosan has been studied for its ability to enhance soil bacterial activity, leading to improved plant growth (*Hallmann et al., 2001*).

However, it was observed that the fungal community in the soil decreased significantly following chitosan foliar spray applications. Previous studies have suggested that this decline in fungal growth might trigger defense responses in plants against pathogens. This could be due to external electrostatic interactions between the positively charged amino glucosamine groups of chitosan and phospholipids in fungal cell membranes, resulting in changes in cell permeability, leakage of intracellular electrolytes, and ultimately cell death (*Divya et al., 2017*; *Ma, Garrido-Maestu & Jeong, 2017*).

Common bean (*Phaseolus vulgaris* L.) is known to be sensitive to salt stress, with significant reductions observed in its morphological attributes and yield when soil salinity exceeds 1.0 dS m$^{-1}$ (*Garcia et al., 2019*). This reduction in yield can be attributed to oxidative damage caused by salt stress (*El-Beltagi et al., 2023*). Our findings corroborate this, as we observed lower values for morphological attributes, seed yield, and NPK content in common bean plants during the growing seasons, which can be attributed to soil salinity leading to high Na concentrations in the plants.

The use of different soil mulching materials resulted in enhancements in chlorophyll content, plant height, fresh weight of shoot and root, seed yield, and NPK content in plants. This improvement can be attributed to the enhancement of soil parameters, including a decrease in soil salinity and an increase in soil fertility and moisture availability. These factors contribute to greater NPK absorption from the soil and lower Na content in common bean plants (*Kwambe et al., 2015*; *El-Wahed et al., 2017*).

Chitosan foliar spray applications have been shown to significantly improve morphological attributes and seed yield while alleviating the adverse effects of salinity stress, as reported by *Krupa-Małkiewicz & Fornal (2018)*. This improvement can be attributed to the action of chitosan, which is known to be more effective against salt stress by reducing oxygen free radicals or blocking ROS activity, promoting cell division, increasing ionic transport, polyamine content, and membrane stabilization under stress conditions (*Balusamy et al., 2022*). Chitosan also enhances water availability to plant and nutrient uptake by adjusting cell osmotic pressure and reducing harmful accumulation of free radicals through increased antioxidants and enzyme activities (*Hidangmayum & Dwivedi, 2022*).

Studies by *Safikhan et al. (2018)* have indicated that the application of nano-chitosan enhances salt tolerance in plants by slowing down chlorophyll reduction, increasing catalase, ascorbate peroxidase, and glutathione reductase activities, influencing the expression of mitogen-activated protein kinases, promoting geissoschizine synthesis, and reducing malondialdehyde levels. Previous research on the role of nano chitosan in salt-stressed common beans has shown improvements in chlorophyll content and plant metabolism, as evidenced by reductions in malondialdehyde and $H_2O_2$ contents (*Sen et al., 2020*), increased leaf area and root length, elevated chlorophyll levels, and carotenoid content (*Santo Pereira et al., 2017*), as well as increased plant height and shoot dry weights due to enhanced antioxidant enzymes (*Zayed et al., 2017*). *Alenazi et al. (2024)* have also demonstrated that nano chitosan has a more beneficial effect on common beans compared

to normal chitosan, whereas it enhances yield, photosynthetic pigment fractions, total carbohydrate content, and the antioxidant system and biochemical traits of common beans.

## CONCLUSIONS

In summary, this study has demonstrated the effectiveness of soil mulching materials in inhibiting salt accumulation and improving saline soil quality. Additionally, the foliar application of 62.5 mg $L^{-1}$ of nano chitosan at 15, 30, and 45 days after sowing (DAS) mitigated salt stress and enhanced the growth of common bean plants. For the promotion of a sustainable ecosystem, the combined application of soil organic mulching materials with foliar chitosan spray can be an effective practice for inhibiting salt accumulation and improving common bean plants cultivated in saline soil. However, future research should address concerns regarding the impact of nano chitosan on soil microbial communities. To further our understanding of the impacts of mulching and chitosan, long-term studies are required to monitor soil microbial communities, including fungi and actinomycetes, to determine the turnover time and rate of the soil microbial community.

## ACKNOWLEDGEMENTS

The authors express their gratitude to the Soil Improvement and Conservation and the Microbiology Labs at the Sakha Agriculture Research Station in Kafr El-Sheikh, Egypt, providing lysimeters and assistance in conducting this research. Additionally, the authors extend their thanks to the anonymous reviewers for their valuable comments and suggestions, which greatly contributed to improving the quality of this study.

### Funding
The Deanship of Research and Graduate Studies at King Khalid University funded this work through Large Research Project under grant number RGP2/118/45. The funders had no role in study design, data collection and analysis, decision to publish, or preparation of the manuscript.

### Grant Disclosures
The following grant information was disclosed by the authors:
Deanship of Research and Graduate Studies at King Khalid University Large Research Project: RGP2/118/45.

### Competing Interests
The authors declare there are no competing interests.

### Author Contributions
- Tamer Khalifa conceived and designed the experiments, performed the experiments, analyzed the data, prepared figures and/or tables, authored or reviewed drafts of the article, materials, analysis tools, supervision, and approved the final draft.

- Nasser Ibrahim Abdel-Kader conceived and designed the experiments, performed the experiments, analyzed the data, prepared figures and/or tables, authored or reviewed drafts of the article, supervision, and approved the final draft.
- Mohssen Elbagory performed the experiments, analyzed the data, prepared figures and/or tables, authored or reviewed drafts of the article, and approved the final draft.
- Mohamed ElSayed Ahmed conceived and designed the experiments, performed the experiments, analyzed the data, prepared figures and/or tables, authored or reviewed drafts of the article, supervision, and approved the final draft.
- Esraa Ahmed Saber conceived and designed the experiments, performed the experiments, analyzed the data, prepared figures and/or tables, authored or reviewed drafts of the article, and approved the final draft.
- Alaa El-Dein Omara performed the experiments, analyzed the data, prepared figures and/or tables, authored or reviewed drafts of the article, and approved the final draft.
- Rehab Mohamed Mahdy conceived and designed the experiments, performed the experiments, analyzed the data, prepared figures and/or tables, authored or reviewed drafts of the article, and approved the final draft.

## Data Availability

The raw data are available in the Supplemental Files.

## Supplemental Information

Supplemental information for this article can be found online at http://dx.doi.org/10.7717/peerj.17828#supplemental-information.

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
