# Peer review of "Investigating the influence of eco-friendly approaches on saline soil traits and growth of common bean plants (Phaseolus vulgaris L.)"

_PeerJ, doi:10.7717/peerj.17828_

## Round 0.1 · original submission · Major Revisions

· Academic Editor

Major Revisions

Dear Dr. Khalifa,

Thank you for your submission to PeerJ.

It is my opinion as the Academic Editor for your article - Effect of eco-friendly strategies on saline soil characteristics and morphological attributes of common bean plants (Phaseolus vulgaris L.) - that it requires some revision.

You are advised to go through all the comments/suggestions by the reviewers and revise your manuscript to address the shortcomings. You will have to take due care with regard to the Materials and Methods part of the study including the experimental design adopted and the experimental procedures used. Furthermore, due emphasis is also required to ensure coherence among different sections of the manuscript.

It is important to mention that your revised manuscript will undergo additional reviews in order to ensure that you have appropriately modified the manuscript, keeping reviewers' suggestions in mind.

Hope to receive the revised submission in due course.

Reviewer 1 ·

Basic reporting

In abstract part
1. The tile is justified according to the study, but it is better to reframe with simple one.
2. Write the full form of EC in the abstract.
3. There are no keywords present in the MS. Kindly write down the keywords and it should be written in alphabetically order.
In introduction part
1. Brief information regarding the plant morphological characteristics, its uses is needed to be mention. Kindly add with citation.
2. In the line number 51, the reference is not mentioned in the references section.
3. In the line number 86, it should be "Divya & Jisha, 2018" instead of “Divya et al., 2018”. Kingly check the references throughout the MS and correct it.

Experimental design

In materials and method
1. In the line number 150, check the years of both the references and write the correct year because in reference section it is mentioned different.
2. In the line number 170, check the name in the reference and correct it.
3. In the line number 188 and 195, the reference is not mentioned in the references section. Kindly mention it and go through the MS and write the references according to the journal’s format.

Validity of the findings

In result part
1. In line number 203-205, how did you get this % of SWD and remaining parameters, kindly mention.
2. In line number 207-208, how did you get this % of RSM kindly mention.
3. In line number 272, you have not mentioned anything as “a, b, c, d” in figure 5, like what "a, b, c and d" is? Kindly check and mention it.
Discussion
1. In line number 320, with same name and year there is two references cited, kindly write "a" and "b" next to the year, so that the reviewer can understand.
2. In line number 326 and 328, the reference is not listed in the reference part. Kindly go through the MS and mention it wherever it is missing.
Conclusion
1. Write the future prospective of the current research.

Additional comments

References
1. Kindly go through the references thoroughly as uniformity in some references is missing like in line number 478, full name of the journal is not written, also check the author’s guidelines for the uniform format.
2. In line number 599, 612, the first letter of the words should be in capital.
In Figure part
1. Proper information of the figure needs to be added and correct the figure numbers accordingly. In figure 5, added the caption (a, b, c, d) as it is not mentioned anywhere. The figure mentioned in the MS is difficult to correlate with the MS 5 (a, b, c, d).

Reviewer 2 ·

Basic reporting

Overall the English and grammar are clear but some sentences need more detail to be complete. For example lines 90-97 the sentence is unclear and a run on sentence without completing the context for clarity. Line 64-68 a different tense of verbs are used. Consider revising the sentences and going through introduction for same tense of sentences and not switching from past tense to present tense helps the reader. The content is organized and literature review relevant some minor English revisions suggested.

The purpose was written well to define the need for best management strategies that incorporate eco-friendly methods to address saline soils and agriculture production of beans.

Thank you for the relevant tables and figures that explain the data but adding more details to explain the figures relevance would help. Please see Zhao et al. 2016 with the sunflowers and plastic mulching as an example. For example figure 5 defines the abbreviations for each mulch type strengthening the figure legend to understand without having to search the manuscript for what Ch 0-3 means and the UNM, WPM, RSM, SDM means. This example also explains the value or standard error.

Experimental design

Materials and methods are mostly clear. A few suggestions to clarify what was intended by the design:

lines 158-160 the description of lysimeter was 48 and size provided earlier which helps but unclear what 2 seeds/hill means and then thinned to a single plant per lysimeter? so 48 plants total?
Line 176 then states 586 samples so just a little unclear of the planting layout.

line 171-173 to expand or be a little more detailed what agriculture practice for beans looks like as current description of practice followed by North Nile Delta region does not mean much unless the reader is from that region.
Line 178 define flag leaf as beans have a leaflet
statistics line 190-195 it's a little unclear of the post-hoc (after ANOVA) testing on how you compared the means. Examples would be Tukey HSD to compare the means (which you did as the letters are in the tables) The methods seems sound but define all post hoc testing.

Validity of the findings

The authors did a great job describing the impact of different mulch treatments and nanotechnology bio-stimulants to aid in saline soils and growth of bean plants in Egypt. The rationale matches the data and findings and support with literature.

Additional comments

Clarifying the methods and a few English grammar ensures an international audience can clearly understand the subject matter and how the project was done. I do feel it is minor edits to revise for clarity and not major for rejection.

Thank you for allowing me to review this interesting research on improving conditions for agriculture.

Reviewer 3 ·

Basic reporting

1. The full form of abbreviation should be mentioned prior to its use. The EC in abstract needs to be mentioned prior to its use.
2. Line no. 90 needs to be redrafted to sync with the flow.
3. In line no. 141, it should be mentioned as foliar spay instaed of four chitosan foliar spay.
4. The a, b, c, d in Table 2 &4 needs to be mentioned.
5. Moreover, the significant codes needs to be made clear specially pertaining to '' 1.
6. The conclusion section needs to incorporate all the present findings as well as give an insight into its future prospect.

Experimental design

The experimental required design is adequate.

Validity of the findings

1. The result of figure 9 needs to be elaborated more.

---

## Round 0.2 · Minor Revisions

· Academic Editor

Minor Revisions

Dear Dr. Khalifa and Dr. Abdelkader,

Thank you for your submission to PeerJ.

Please rectify the footnote/figure issue and resubmit ASAP.

Reviewer 1 ·

Basic reporting

Dear Author,
I have carefully re-reviewed the article “Investigating the Influence of Eco-Friendly Approaches on Saline Soil Traits and Growth of Common Bean Plants (Phaseolus vulgaris L.)” were the author had rectified accordingly however, the figure number doesn’t align with the footnote kindly go through it.

Experimental design

the figure number doesn’t align with the footnote kindly go through it.

Validity of the findings

the figure number doesn’t align with the footnote kindly go through it.

Additional comments

the figure number doesn’t align with the footnote kindly go through it.

---

## Round 0.3 · accepted · Accept

· Academic Editor

Accept

Dear Dr. Khalifa and Dr. Abdelkader,

Thank you for your submission to PeerJ.

I am writing to inform you that your manuscript - Effect of eco-friendly strategies on saline soil characteristics and morphological attributes of common bean plants (Phaseolus vulgaris L.) - has been Accepted for publication.

Congratulations!

Reviewer 1 ·

Basic reporting

Dear Author,
I have carefully reviewed the article “Effect of eco-friendly strategies on saline soil characteristics and morphological attributes of common bean plants (Phaseolus vulgaris L.)” where all the previous comments has been rectified and the MS is ready to submit.

Experimental design

Dear Author,
I have carefully reviewed the article “Effect of eco-friendly strategies on saline soil characteristics and morphological attributes of common bean plants (Phaseolus vulgaris L.)” where all the previous comments has been rectified and the MS is ready to submit.

Validity of the findings

Dear Author,
I have carefully reviewed the article “Effect of eco-friendly strategies on saline soil characteristics and morphological attributes of common bean plants (Phaseolus vulgaris L.)” where all the previous comments has been rectified and the MS is ready to submit.

Additional comments

Dear Author,
I have carefully reviewed the article “Effect of eco-friendly strategies on saline soil characteristics and morphological attributes of common bean plants (Phaseolus vulgaris L.)” where all the previous comments has been rectified and the MS is ready to submit.